# Online Resources for People Who Self-Harm and Those Involved in Their Informal and Formal Care: Observational Study with Content Analysis

**DOI:** 10.3390/ijerph17103532

**Published:** 2020-05-18

**Authors:** Daniel Romeu, Elspeth Guthrie, Cathy Brennan, Kate Farley, Allan House

**Affiliations:** Leeds Institute of Health Sciences, University of Leeds, Leeds LS2 9NL, UK; danielromeu@doctors.org.uk (D.R.); c.a.brennan@leeds.ac.uk (C.B.); k.farley@leeds.ac.uk (K.F.); a.o.house@leeds.ac.uk (A.H.)

**Keywords:** self-harm, internet, self-help, advice, guidance

## Abstract

Despite recent fears about online influences on self-harm, the internet has potential to be a useful resource, and people who self-harm commonly use it to seek advice and support. Our aim was to identify and describe UK-generated internet resources for people who self-harm, their friends or families, in an observational study of information available to people who search the internet for help and guidance. The different types of advice from different websites were grouped according to thematic analysis. We found a large amount of advice and guidance regarding the management of self-harm. The most detailed and practical advice, however, was limited to a small number of non-statutory sites. A lay person or health professional who searches the web may have to search through many different websites to find practical help. Our findings therefore provide a useful starting point for clinicians who wish to provide some guidance for their patients about internet use. Websites change over time and the internet is in constant flux, so the websites that we identified would need to be reviewed before making any recommendations to patients or their families or friends.

## 1. Introduction

There are over 220,000 hospital attendances for self-harm per annum in the UK and approximately 100,000 acute admissions [1]. In recent years, the prevalence of self-reported self-harm has greatly increased, particularly in young women [2]. The estimated annual rates of presentation of self-harm in primary care are 18.5 and 28.9 per 10,000 male and female patients [3], respectively, so any moderate to large practice would expect to see 40 or more cases of self-harm per annum.

The internet is now ubiquitous globally and used for social, occupational, educational and recreational purposes. A 2015 study of people who self-harmed reported that 22% of its participants had used the internet to search for sites related to self-harm or suicide [4]. Most people who self-harm use the web for constructive reasons, such as finding information or to participate in forums [5,6]. Concern has been raised about on-line use which may encourage or normalise self-harm or suicide, particularly in young people [6], and there has been a high-profile discussion about this in the media in the UK [7,8], with a focus on sites that show images of self-harm or involve some form of cyber bullying, or sites that are said to normalise self-harm so that they prevent individuals from seeking help [9,10,11]. The recent report by the Royal College of Psychiatrists on ‘Technology Use and the Mental Health of Children and Young People’, supported this perspective by including a forward by a parent of a young adult who died by suicide which stated, ‘I have no doubt my social media helped kill by daughter’ [7].

An alternative perspective is provided by the observation that people who self-harm are more likely to seek access to self-harm related sites which offer help and advice than sites which provide information on how to hurt oneself [4]. For example, a recent systematic review of the relationship between internet use, self-harm and suicidal behaviour in young people identified 46 studies of relevance [12]. The authors concluded there was the potential for internet use to reduce isolation and to be used as a source of help and therapy, and that clinicians who work or see people who self-harm should engage in discussion about internet use. A study that explored the nature of images tagged as self-harm on three popular social media sites found that none of the images portrayed images of graphic or shocking self-injury, and none could be construed as sensationalising self-harm [13]. The researchers did not find evidence of any posts that could be viewed as actively encouraging others to self-harm. Rather, they concluded that sites were being used mainly to express difficult feelings or offer encouragement to others to stop self-harming.

In the context of this debate, it would be useful for clinicians if guidance were available about which aspects of the internet they should discuss with patients, or where patients or their families and friends might be directed on the web to find help. Such guidance, however, is not readily found. We have been able to identify only one previous study conducted in 2001, which sought to describe the information and help available on the internet for people who self-harm, and aimed at informing clinicians about potential resources when discussing self-harm with their patients [14]. The authors found only a small number of citations (31–37) with content about various aspects of self-harm. Most of the cites provided constructive suggestions (e.g., how to overcome shameful feelings about self-harm, or advice on how to feel better about oneself). Most of the information concerned self-injury and there was very little discussion of self-poisoning.

The aim of the present study was to update the 2001 study to investigate, characterise and describe the resources currently available on the internet for people who self-harm, or their friends or families, which may be potentially helpful. We focused on UK-generated sites, so that our findings could be used by UK clinicians to guide their own searches and subsequent discussions with their patients. We excluded websites that required log-in or registration, or on-line forums, as our focus was specifically on sites that might contain direct and easily accessible guidance.

## 2. Materials and Methods

Targeted web searches were conducted between 18th August and 26th September 2019 by the lead author (D.R.). An initial scoping exercise identified UK based websites of public bodies, professional health organisations, Royal Colleges, therapy and counselling organisations, public health agencies and support groups that may potentially provide information on self-harm and mental health. This involved a combination of web searches, consultations with experts in self-harm (A.H. and C.B.) and clinicians involved in self-harm services. Identified sites were searched for references/citations to other sites, which were then, in turn, searched for other references to sites that have not been included in the initial searches. Forty-nine United Kingdom (UK) organisations were identified (shown in Appendix A). The entirety of each of the 49 websites, starting with its home page, was browsed for pages and for documents either embedded or accessible through hyperlinks. Individual searches were conducted for each site using each of the following search terms in quotation marks: “self-harm”, “deliberate self-harm”, “self-injurious behaviour”, “self-injury”, “self-cutting”, “self-poisoning” and “parasuicide”. This strategy yielded 4988 results in the form of webpages and documents.

A customised, advanced Google search was conducted on 26th September 2019. This complementary searching strategy was incorporated to maximise the likelihood of including key, relevant sources and information, and to emulate what a service user or health professional would be likely to find if seeking help in managing self-harm. The following search terms were used: “self-harm” OR “self-injury” OR “self-injurious behaviour” OR “deliberate self-harm” OR “self-cutting” OR “self-poisoning” OR “parasuicide”. The results were filtered to include results written in the English language and from the United Kingdom. The first 100 search results were screened to capture the most relevant hits, while still representing a feasible number to screen.

The results of both the targeted website searches and the advanced Google search were screened for eligibility using the website title, table of contents, overview, executive summary, or whatever other information was available. The eligibility criteria are shown in Table 1.

When it was unclear whether the item met the eligibility criteria, the item was included for full-text screening. As the targeted website search strategy differed slightly between websites, due to different search and filtering options available, the precise search strategy and results for each website can be viewed in Appendix A, as well as further relevant webpages identified by analysing the targeted search results. Of the advanced Google search results, 55 were included for full-text analysis; reasons for exclusion can be viewed in Appendix A.

The full text of each search result was read by one researcher for any practical help or guidance aimed at helping reduce self-harm. Where possible, policy documents, guidelines and quality and competency frameworks were downloaded for analysis. Data were extracted pertaining to the organisation, date accessed, intended audience and the goals of the resource. Extracts from the search results that were deemed relevant were recorded verbatim in a summary table on Microsoft Word.

Given the exploratory nature of the review, and the principal aim to identify themes directly from the data rather than a priori, inductive thematic content analysis was selected. Further merits of this approach include its flexibility and lack of commitment to a pre-existing theoretical framework [15]. This process was assisted by the use of the software package NVivo 12 Plus (QRS International, Burlington, MA, USA).

Two researchers read and re-read the dataset individually (DR is a male junior doctor; EG is a female academic psychiatrist) and independently coded meaningful units of text within the data. The codes were then reviewed and compared, and any discrepancies were resolved by consensus. The codes were collaboratively moved into overarching themes, relevant to the research question. Themes were checked in relation to the data extracts to ensure accurate representation of the data, and to generate a thematic map consisting of meta-themes, themes and sub-themes.

As both people carrying out the analysis were medically trained, it is possible that the thematic analysis may have been influenced by a medical bias in terms of developing themes. To ensure intersubjective reproducibility and comprehensibility the analysis was regularly discussed with a multi-disciplinary team of professionals including self-harm experts (A.H. &C.B.), and qualitative researchers. The dataset consisted of all factual information, which was available on each site regarding self-harm. Any value judgement attached to that information (e.g., whether it was likely to be helpful or unhelpful) was discussed within the team, and then further discussed within the setting of the workshop, which included a diverse range of nine people with lived experience of self-harm, in addition to two self-harm experts, three clinical psychologists, two psychiatrists, one junior doctor and two mental health nurses.

The dataset was re-visited at pertinent points throughout the analysis. Direct quotations have been selected and incorporated to both illustrate the themes identified in the literature and demonstrate the breadth in content. Consolidated Criteria for Reporting Qualitative Research (COREQ) criteria [16] for reporting qualitative research were followed throughout the stages of data analysis and synthesis, where appropriate.

## 3. Results

### 3.1. Searches

The targeted website searches identified 4988 webpages or documents, with 383 ultimately meeting the inclusion criteria for full-text inclusion and analysis. 2380 were duplicates, and 2225 did not meet eligibility criteria. The search strategy, number of results and duplicates and details of results included for each constituent website can be found in Appendix A. Each of the 49 websites are listed with the number of hits for self-harm, duplicates and the number of hits that met eligibility criteria for full text inclusion e.g., the website for the National Institute for Health and Care Excellence (NICE) had 82 hits (16 duplicates), of which 33 met eligibility for full text inclusion. The 49 sites generated 2694 hits (excluding duplicates), of which 383 hits met eligibility for full text inclusion. Of the 100 advanced Google search results, fifty-six were included for analysis, and forty-four were excluded; further details can be viewed in Appendix A. The study flow diagram according to PRISMA guidelines is shown in Figure 1 [17].

We found considerable diversity in the type of information provided regarding target audience and the nature of the material. We also found several websites that focused on specific guidance for prisoners who self-harm, people with learning difficulties and young people, the content of which will be discussed later.

We identified the following groups:

#### 3.1.1. Help Offered by Professionals

Talking therapy was recommended by twenty-four of the forty-nine targeted websites. A variety of approaches were highlighted, including cognitive-behavioural therapy, dialectical behavioural therapy and psychodynamic therapies e.g.,


*“There is evidence that some ‘talking therapies’, e.g., problem-solving therapy, psychodynamic therapy and cognitive behavioural therapy (CBT) are useful in helping people who harm themselves.”*
(Improving the lives of people who self-harm, HSC Public Health Agency)

The recommended duration of therapy varied amongst sources, although three to twelve sessions was suggested on multiple websites. Little information was provided about the nature of different therapeutic approaches or their mechanism of action, or how people could access help and its likely availability. A positive relationship between the client and the therapist was widely acknowledged as being important in reducing self-harm, with the importance that the therapist was non-judgmental, caring and sympathetic.


*“My relationships with the psychotherapist and the psychologist were a lifeline, and my psychotherapist certainly kept me alive many times.”*
(Northern Ireland, United Kingdom: Caitriona Cassidy, World Health Organisation)

Other forms of help included the recommendation of developing a risk management plan, which was cited by four professional bodies.


*“A risk management plan can help people who self-harm reduce their risk of self-harming again. It should be based on a risk assessment and developed with the person who has self-harmed, who should have joint ownership of the plan.”*
(Self-harm [QS34], NICE)

Other guidance highlighted the importance of inter-professional team working and the role of the GP in either initiating referral to psychological services, or being included in risk management plans.


*“Primary care practitioners, in applying the recommend bio-psycho-social approach, should ensure prompt referrals to psychology (IAPT) services.”*
(Mental health: self-harming in older adults has distinct characteristics, Royal College of General Practitioners)


*“Summarise the key areas of needs and risks identified in the assessment and use these to develop a care plan and a risk management plan in conjunction with the person who self-harms and their family, carers or significant others if this is agreed with the person. Provide printed copies for the service user and share them with the GP.”*
(Self-harm in over 8s: long-term management [CG133], NICE)

#### 3.1.2. Help Offered by Non-Professionals

Fourteen targeted websites and six Google results advised that talking to others is useful in the management of self-harm:


*“Lots of young people have said that telling someone about their self-harm was one of the best ways of coping.”*
(Self-harm, Childline)

As with professionals, a non-judgmental approach was advised, as well as being patient, kind and respectful.


*“If you can, try and stay calm and non-judgemental – it will help both you and your friend.”*
(How to react when your friend says they self-injure, LifeSIGNS)


*“[People who self-harm] often report that the sympathy, tolerance and respect of those close to them is integral to getting their self-harm under control.”*
(Self-harm – what, who, why and how to help, British Psychological Society, identified via Self Injury Support Network)

The development of social networks, either specifically related to self-harm support or involving other activities was recommended by several sites, and the role of the family was seen as being a very positive potential vehicle for change, although of course this is not always possible.


*“A group of people, who all self-harm, meet regularly to give each other emotional support and practical advice. Just sharing your problems in a group can help you to feel less alone - others in the group will almost certainly have had similar experiences.”*
(Self harm, Royal College of Psychiatrists)


*“Sometimes joining a social activity or sports group can be helpful as a distraction. This can also provide a form of social support.”*
(Coping with self-harm; Royal College of General Practitioners)


*“Mummé and her colleagues report that family support was the “predominant interpersonal” factor associated with stopping self-harming, including in studies that involved adults, not just those with teens and children.”*
(Family support in self-harming, British Psychological Society)

A variety of useful sources of support for those struggling with self-harm were identified, including: applications; supportive on-line communities; and positive accounts of recovery from self-harm.


*“Seek out online communities of people who are honestly looking to support each other overcome self-harm. Support from others is incredibly powerful and can help you refrain from hurting yourself.”*
(We Need To Talk About Online Self-Harm Content, The Mix)


*“Reboot your social media by following different, positive accounts. By doing this you’ll fill your feed with positive content, coping strategies, images and messages that are going to raise you up rather than beat you down.”*
(We Need To Talk About Online Self-Harm Content, The Mix)

### 3.2. Strategies to Help Stop Self-Harming

A variety of websites offered specific advice to help people at imminent risk of an act of self-harm. The most common strategies are listed in Table 2. More general advice was provided about changes that people could make to reduce the possibility of experiencing thoughts of self-harm as opposed to what actually to do when the urge to self-harm was present. These are shown in Table 3. The advice was wide ranging, including general lifestyle changes and more specific advice about dealing with underlying issues driving self-harm.

### 3.3. Personal Accounts of Recovery

There were several personal accounts of recovery, as well as reference to studies that had interviewed people with self-harm about the recovery process. We identified four themes:

#### 3.3.1. Self-Discovery and Development, whether Alone or through Help from Psychological Therapy

*“The start of my healing came* via *creative writing. I was too ashamed and afraid to talk about what was happening so I put it onto paper instead.”*(Terrible times, self-injury and recovery, LifeSIGNS)


*“Other important factors that enabled people to stop harming included self-development, sometimes in the form of insight into reasons for harming gained through counselling, other times simply through growing older and benefiting from the increase in confidence, stability, self-knowledge and control over life associated with this.”*
(Self-harm, SANE)

#### 3.3.2. Contemplation about and Motivation to Stop Self-Harming


*“An important step forward is making the decision to learn to live without self-harm, and being prepared to face underlying issues that may have caused the behaviours in the first place.”*
(Recovering from self-harm, SelfHarmUK)

#### 3.3.3. The Impact of Self-Harm on Others


*“My parents found out and made me feel bad about it… I’m glad I don’t do it anymore but only because I don’t want to upset anyone.”*
(Self-Harm, SANE)


*“Interestingly, one study found that [people who used to self-harm] saw their self-harm as a useful coping mechanism, but had been motivated to stop because their loved ones wanted them to stop.”*
(Family support in self-harming, British Psychological Society)

#### 3.3.4. The Negative Impact of Scars


*“Another thing that made me break away from self-harming is the fact that my scars started to make me feel even more insecure than I already was.”*
(Self-harm is real, SelfHarmUK)

### 3.4. Specific Demographic Groups

Some sources focused on self-injury in people with autism and special educational needs. Unique approaches to managing self-harm in this group were identified. For example, professional bodies recommended antipsychotics in particular circumstances or use of a diary by a carer to record and identify the immediate precipitants of the self-harming:


*“Complete a behaviour diary, which records what is occurring before, during and after the behaviour. This could help you to understand the purpose of the behaviour. Makes notes on the environment, including who was there, any change in the environment and how you think the person was feeling.”*
(Self-injurious behaviour, National Autistic Society)

Five of the forty-nine websites discussed self-harm amongst prisoners. Various specific interventions were recommended, including an Assessment, Care in Custody and Teamwork (ACCT) plan and the Listener scheme:


*“ACCT is a prisoner-centred, flexible care-planning system which, when used effectively, can reduce risk, primarily of self-harm.”*
(Mental health of adults in contact with the criminal justice system [NG66], NICE)


*“The Listener scheme is a peer supportive service which aims to reduce suicide and self-harm in prisons. Prisoners are trained and supported by the Samaritans to become Listeners. Listeners can provide confidential emotional support to you when you are struggling to cope.”*
(Prisoners and Self Harm, Rethink Mental Illness)

Some websites particularly focused on young people, with age-specific advice, which involved advice about access to the smart devices, involvement of schools and other youth organisations, or specific treatment guidance, which differed from adults.


*“RCPsych recommends that children stop using technology at least an hour before going to bed and avoid using technology at mealtimes.”*
(Psychiatrists should consider impact of social media on all children they assess, Royal College of Psychiatrists)


*“Find out who the Designated Safeguarding Lead Person is for the young person’s school and arrange to have a meeting with them so that you can address the situation. They will be able to help you put an action plan in place to keep the young person safe, and will explore ways you can work together to bring an end to the self-harming behaviour.”*
(Understanding Self-Harm in Young People, Child Protection Company)

### 3.5. Website Review

We noted that three sites, LifeSIGNS, Self Injury Support Network and The Mix, provided over thirty relevant webpages with detailed, practical advice to help people stop self-harming or cope better. SelfHarmUK, Recover Your Life, Young Minds and Mind also provided direct advice to the reader about self-harm.

Brief professional advice was offered by the World Health Organisation, National Institute for Health and Care Excellence, Royal College of Psychiatrists, British Association for Counselling and Psychotherapy, British Psychological Society and Public Health England. The Centre for Mental Health summarised research findings in the field, relevant to both professionals and academics.

## 4. Discussion

There has been a large increase in the amount of advice and guidance on the web about self-harm since the study in 2001 by Prasad and Owens [14]. The most common guidance suggested that talking to others about self-harm was an important step to recovery, either via professional talking therapies or confiding in friends or family. The importance of a non-judgmental and supportive approach by therapists, health professionals or family and friends was also highlighted. A range of suggestions was available about methods to stop actual episodes of self-harm and most of the sites that covered this area included broadly overlapping strategies. Many sites also focused on the need for personal change to address the underlying reasons for self-harm or to improve overall wellbeing. Individual accounts of recovery focused on the need for self-determination and motivation to stop self-harming.

Most information, however, is focused on cutting, and there is still relatively little specific help that addresses self-poisoning. Much more information is available for specific demographic groups of people than in 2001 and there are many more personal accounts of recovery. A recent analysis of YouTube videos that described recovery from self-harm identified several themes which overlap with the ones we identified in this study: encouraging help seeking and talking about problems; substituting self-injurious practices by using alternative methods; a desire for change (motivation); and control over change (self-determination) [18].

The most detailed and practical advice, however, was limited to a relatively small number of non-statutory sites, so a lay person or health professional who searches the web for help about self-harm may have to search through many different websites to find this useful guidance and help.

A recent study reported that almost one quarter of young adults in the community had accessed Internet-content related to suicide/self-harm [4], and almost half of those who had self-harmed had searched for information about suicide or self-harm. A greater proportion of young people had accessed helpful sites than potentially harmful sites. Ten per cent of the whole sample of 3946 participants had searched for general information about self-harm, 8.2% for sites offering help, advice or support, and 9.1% for personal accounts of people who had hurt themselves. A smaller proportion (4%) had accessed sites dedicated to those who self-harm.

Another study of people attending hospital who had self-harmed also found high levels of internet use, particularly among children [19]. It is therefore important that health professionals are aware of this, and feel comfortable discussing Internet use with the people they are trying to help. It has been suggested that clinicians should take an Internet history during psychosocial risk assessments for self-harm [20], and generally, this seems to be something that clinicians feel is acceptable to do [19], although it is not known how common this is in clinical practice.

Strengths of this study are the dual independent coding and discussions of the analyses in a multidisciplinary workshop, including psychiatrists, clinical psychologists, experts in self-harm, mental health nurses and people with lived experience of self-harm. There are some limitations. We limited our search of targeted websites to UK bodies and organisations as we sought to discover those aspects of advice (such as contact details for statutory and non-statutory organisations) that were relevant to UK citizens. Self-harm is a world-wide problem, and different professional and health organisations will vary from country to country. The basic guidance that we found is applicable to most people and settings, but inevitably there will be social and cultural differences between countries and organisation of statutory and non-statutory services for people who self-harm. The main analysis was undertaken by two researchers from a clinical background and did not involve people with personal experience. However, our findings were reviewed in a workshop attended by people with personal experience, who were supportive of our conclusions. The search terms we used were generated within the research group by discussion and then tested out with preliminary searches to determine what sites they found or other terms were revealed. We realise that some other common terms such as ‘overdose’ were not included in the terms we generated.

We used common terms for self-harm in our search of websites and Google, as opposed to a comprehensive search strategy, as we wanted to mimic searches that an ordinary individual might make. Inevitably, this means we may have missed some websites or information about self-harm. We did not explore information that is provided in chat rooms, or on closed websites requiring a membership, as we sought to discover what is freely available on the web to people who self-harm, their family and friends and health professionals.

### Clinical and Research Implications

Our findings provide a starting point for clinicians who wish to provide some guidance for their patients about internet help resources for self-harm. NICE guidelines highlight the important role of primary care in the assessment and treatment of self-harm [21], thus, our findings are likely to have useful applications in a primary care setting.

Most of the advice that we found was quite basic and may be helpful for people with little or no personal experience of self-harm, but is unlikely to be helpful for people who have been self-harming for some time. Any discussion or reference to particular sites should not come with a recommendation that they are good, but more that the person may find them of some use.

Our findings provide a starting point for clinicians who wish to provide some guidance for their patients about internet help resources for self-harm. A recent editorial on the challenges for general practice in managing self-harm in young people suggested discussion and use of internet resources as part of a range of potential helpful resources people may access [22]. We suggest that clinical trials to evaluate psychosocial interventions for self-harm should record the nature of Internet use by trial participants.

## 5. Conclusions

We have undertaken a review of the considerable amount of advice available on the internet for people who self-harm. We have found no similar up to date summary of online resources. Websites change over time and the internet is in constant flux, so the websites that we identified which provide detailed advice would need to be reviewed, before any recommendations to patients or their families or friends.

Enquiry about internet use should be included in clinical assessments of people at risk of self-harm, especially young people. The Royal College of Psychiatrists guidance to clinicians emphasises the need to explore possible harms, so a fuller discussion should be supplemented with questions about what has been found helpful, and with advice about sites or organisations that can be readily accessed. Such advice should always come with a comment that what is helpful varies by individual, and with an offer to discuss experiences again if wanted.

## Figures and Tables

**Figure 1 ijerph-17-03532-f001:**
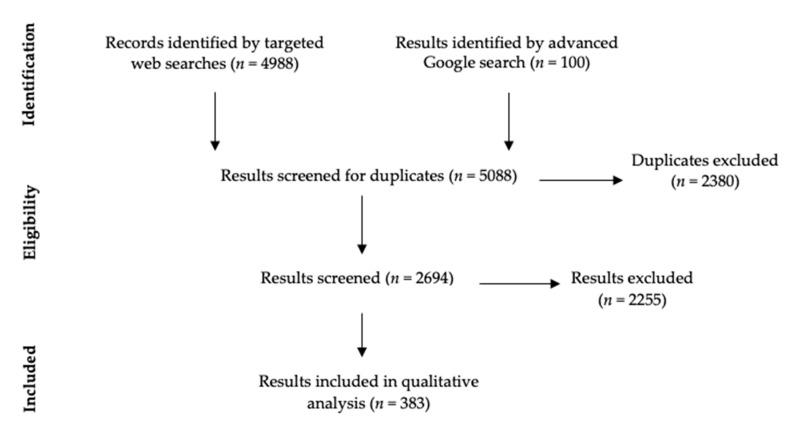
PRISMA Flow Diagram.

**Table 1 ijerph-17-03532-t001:** Grey literature review eligibility criteria.

Inclusion Criteria	Exclusion Criteria
Literature that Provides Information or Advice about what Helps to Stop or Reduce Self-Harming Behaviour Available in English Published by an Organisation within the United Kingdom Literature Related to or Aimed at Individuals with or without Co-Occurring Psychiatric Disorders Literature Across all Motives (Non-Suicidal or Suicidal) and Methods (Poisoning or Self-Injury) of Self-Harm	Empirical studies Online forums Social media Inaccessible to the public (page not available, requires log-in)

**Table 2 ijerph-17-03532-t002:** Strategies to help avoid imminent self-harm.

Strategy	Number of Websites	Examples
Delaying (The 5 Minute Rule)	9	“Some people find that putting off harming themselves can decrease or get rid of the urge.” (Coping with self-harm: a guide for parents and carers, Royal College of Psychiatrists) “If you find distraction doesn’t work, you could also try delaying. This means waiting for five minutes before you self-harm and see if the urge is as strong. Try to build up the amount of time you can put it off.” (Self-harm, Counselling Directory)
Distraction	12	“Distraction techniques can help you resist the urge to self-harm until the feeling passes” (Self-harm Rape Crisis, identified via Self Injury Support Network) Some guidance grouped distraction methods around different emotions to suggest ways that some techniques, such as writing or running, might be more effective depending on what might be behind an urge to self-harm”. (What to do if you get the urge to self-harm, Patient) Examples of distraction techniques were: Talking to someone you trust Writing letters Cleaning and tidying Exercise Reading a book Listening to music
Substitution (Replacement of the Act of SH with Other Acts that Satistfy the Urge)	11	“Replacing the cutting or other self -harm with safer activities can be a positive way of coping with the tension.” (Supporting Children and Young People who Self-Harm, Northamptonshire Children and Young People’s Partnership, identified via Self Injury Support Network) “Try to satisfy that urge without hurting yourself. It can be anything from flicking elastic bands against your wrist, rubbing ice cubes on your skin, or dropping hot sauce on your tongue.” (Self harm is an addictive way to cope with mental pain, The Mix)
Harm Minimisation	4	“If stopping self-harm is unrealistic in the short term, consider strategies aimed at harm reduction.” (Self-harm in over 8s: long-term management [CG133], NICE)
Hiding Objects	4	“Reducing the accessibility of objects that might be used for self-harm (e.g., pencil sharpeners, knives, medication etc.) may help to delay the impulse to self-harm.” (Coping with self-harm: a guide for parents and carers, Royal College of General Practitioners)
Relaxation Techniques	7	“Try relaxation and breathing exercises. Sit back comfortably in a chair or lie out on a bed. Relax all muscles in your body, beginning at the feet and working upwards. Concentrate on your breathing: breathe in for 5 s through your nose, hold your breath for 5 s, then breathe out slowly. Repeat this.” (Self-harm, Survivors Manchester, identified via SurvivorsUK)
Identifying and Avoiding Immediate Precipitants (E.G. Alcohol, or Looking at Websites that May Cause Distress)	11	“Identifying what triggers your self-harming can give you more control. Even if you can’t avoid those triggers altogether, you can develop strategies to deal with your emotions when things start becoming overwhelming.” (How to get control over your self-harm, SelfHarmUK)
Acknowledging Choice	5	“All behaviours are a choice, and if we stop to take a breath before self-injuring, and recognise that we are responsible for the continuation of our destructive cycles, then we become empowered to break them.” (Choice, LifeSIGNS)
Mindfulness	4	“Many people find mindfulness techniques very helpful in getting past the urge to self-injure.” (Self-Injury Awareness Day, Mental Health Matters)
Go to a Safe Place	7	“Most people self-harm when alone so go to a public place, be with a good friend or a safe family member. This may prevent you from harming yourself.” (Self-harm, a self-help guide, Survivors Manchester, identified via SurvivorsUK)
Planning Ahead	10	“Make a self-soothe box – work with the young person to collect a range of different things they can use to distract or soothe themselves when they feel the urge to self-harm. This might include music, colouring, books, bubbles, photographs or inspirational quotes.” (No Harm Done, Young Minds) “You may also find it helpful to write a list of all the people, organisations and websites that you can go to for help when you are finding things difficult.” (Helping yourself long-term, Mind)

**Table 3 ijerph-17-03532-t003:** Ways of reducing thoughts or impulses to self-harm.

Strategy	Number of Websites	Examples
Understanding Self-harm	20	“Perhaps the most important thing is to think about why you’ve started to self-harm in the first place and what purpose it serves for you.” (Steps to self-harm recovery, The Mix)
Identifying and Managing Emotions	19	“Try to be clear about what you are feeling – is the emotion you are feeling: fear, shame or guilt, anxiety, anger, rage, sadness or depression? Try and observe, label and accept the emotion. Ask yourself why you are feeling it.” (Self-harm, Survivors Manchester, identified via SurvivorsUK) “Many people who self-harm do so repetitively, because although it is destructive it releases the unbearable tension and therefore it seems to ‘work’. Self-harmers need to learn a different means of expressing emotion”. (Self-harm, what, who, why, and how to help, British Psychological Society, identified via Self Injury Support Network)
Positive Thinking	5	“It may also be helpful to try and encourage the student to think about some positive things about themselves and their life and develop a ‘hope box’ where they can store things that make them feel better, such as photos, memories, nice things people have said etc.” (Young people who self-harm, University of Oxford, identified via Self Injury Support Network)
Mindfulness	4	“Mindfulness may teach you to be more aware of your thoughts and feelings. Once you are more aware of your thoughts and feelings, you can learn to deal with them better.” (Prisoners and Self Harm, Rethink Mental Illness)
Improving General Well Being	10	“When we’re taking care of ourselves and are seeking help and support for the issues that are causing us distress, we feel good that we are helping ourselves and are taking steps along a path of recovery” (Choice, LifeSIGNS)
Developing New Ways of Coping	17	“Moving away from self-injury however requires learning new coping mechanisms that can help you move towards change” (New Year, new you?, LifeSIGNS)
Tackling Underlying Issues	16	“I think the best way to stop self-harm is to focus on the underlying issues which trigger you to do it. If you work on these issues, then the self-harm will stop naturally”. (Understanding self-harm, Scottish Association for Mental Health)
Treating Mental Illness	8	“Untreated mental health problems may contribute to an increased risk of continuing self-harm or even suicide”. (Study looks at self-harm in young people, NHS)
Faith and Religion	1	“It’s hard to explain the awesomeness and the powerfulness of what God did inside me that week. He gently gave me the strength to ask for help, protecting me and showing me how His strength was better than anything I could ever get from self-harming, showing me I was worth a lot more to Him.” (Survivors’ stories, Adullam Ministries)
Developing a Creative Outlet	10	“Do something creative: make a collage of colours to represent your mood or to remind you of your favourite things.” (The truth about self-harm, Mental Health Foundation)
Seek Professional Help	15	“Encourage them, if they have not already done so, to seek professional help. Offer to go with them to their GP or a counsellor. If you’re at school, perhaps there is a teacher they trust of the school nurse.”. (How to react when your friend says they self-injure, LifeSIGNS)

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
