# Peer review of "Online Resources for People Who Self-Harm and Those Involved in Their Informal and Formal Care: Observational Study with Content Analysis"

_ijerph, 2020, doi:10.3390/ijerph17103532_

Round 1

Reviewer 1 Report

Thank you for the opportunity to review this manuscript. The authors present some interesting findings about online resources for people who self-harm. This topic could be very useful for readers who are trying to understand what sites might be valuable, particularly from a lived experience perspective. However, the manuscript needs development.

The introduction is largely descriptive and quite vague. It tends to list studies rather than argue for the research presented. The authors also assume that many of their points are self-evident. For example, the authors state that the main focus of controversy in the UK has been around online sites that ‘normalise self-harm’. However, the authors provide no information or evidence for this. The final point about updating a previous study is particularly vague. What did the study highlight? Why does it need updating? What do the authors aim to find out? The authors need to make a clear case for a gap in the literature and have a clear aim for their study.

The methods and findings are also rather vague, as well as being hard to follow. For example, P2, line 75, there is a reference to experts being consulted, but it’s not clear who and how many experts were consulted or why. Also, the authors refer to a ‘snow-ball technique’ for identifying sites. Is this a method that has been used previously and successfully in other research of this kind? The search strategy is hard to follow. Perhaps a diagram would help. The analysis was conducted by a medical student and psychiatrist and it’s not clear how this influenced the analytical process. Particularly, what measures were taken to ensure that people with lived experience were able to speak openly about their perspectives, and how were these perspectives privileged or valued? The findings are listed on page 4, but the categories do not reflect this list. The presentation of findings is also inconsistent. Some sections of the findings are presented as a series of quotes and some are presented in tables. In both cases there is very little analysis, just descriptive quotes.

Finally, the discussion doesn’t bring the piece together. For example, the authors make the comment that some sites offered more practical advice, but it’s not clear how they came to this conclusion. There is little to no discussion about peer vs professional advice, which would seem an important consideration. The discussion needs to be more integrated with the wider literature to understand what this study adds, including what it adds to the previous 2001 study.

Author Response

  1. The introduction is largely descriptive and quite vague. It tends to list studies rather than argue for the research presented. The authors also assume that many of their points are self-evident. For example, the authors state that the main focus of controversy in the UK has been around online sites that ‘normalise self-harm’. However, the authors provide no information or evidence for this.

Response: We thank the reviewer for their comments. We do not claim that normalisation of self-harm online is the main focus of controversy, only that it is one concern. We have supported the point we made about controversy in the UK by referring to the report by Dubicka and Theodosiou (Royal College of Psychiatrists, 2020 )which has a forward by the father of Molly Russell, a teenager who committed suicide, in which Mr Russell states, ‘I have no doubt that social media helped kill my daughter’ . The opening sentence of the report states, ‘The use of screen time and social media by young people has rarely been put of the headline in recent years’. We also reference the UK Government’s White Paper on Online Harms.  Lines 42 - 49

“Concern has been raised about on-line use which may encourage or normalise self-harm or suicide, particularly in young people [6] , and there has been a high-profile discussion about this in the media in the UK [7,8], with a focus on sites that show images of self-harm or involve some form of cyber bullying, or sites that are said to normalise self-harm so that they prevent individuals from seeking help [9-11]. The recent report by the Royal College of Psychiatrists on ‘Technology Use and the Mental Health of Children and Young People’, supported this perspective by including a forward by a parent of a young adult who died by suicide which stated, I have no doubt my social media helped kill by daughter’ [7].”

  1. The final point about updating a previous study is particularly vague. What did the study highlight? Why does it need updating? What do the authors aim to find out? The authors need to make a clear case for a gap in the literature and have a clear aim for their study.

Response: We have changed the text to indicate more clearly that the purpose of the study is to identify sites that are likely to be immediately useful to clinicians and others seeking online advice about self-harm and added details which describe the results of the previous study. Lines 62-71

"In the context of this debate, it would be useful for clinicians if guidance were available about which aspects of the internet they should discuss with patients, or where patients or their families and friends might be directed on the web to find help. Such guidance is however not readily found. We have been able to identify only one previous study, conducted in 2001, which sought to describe the information and help available on the internet for people who self-harm – aimed at informing clinicians about potential resources when discussing self-harm with their patients [14]. The authors  found only a small number of cites (31-37) with content about  various aspects of self-harm. Most of the cites provided constructive suggestions (e.g. how to overcome shameful feelings about self-harm, or advice on how to feel better about oneself). Most of the information concerned self-injury and there was very little discussion of self-poisoning."

  1. The methods and findings are also rather vague, as well as being hard to follow. For example, P2, line 75, there is a reference to experts being consulted, but it’s not clear who and how many experts were consulted or why. Also, the authors refer to a ‘snow-ball technique’ for identifying sites. Is this a method that has been used previously and successfully in other research of this kind? The search strategy is hard to follow. Perhaps a diagram would help.

We have clarified that the experts (AH and CB) were part of the research team and have also clarified below why they were consulted (see response to next point). Regarding the term snowballing- we have removed  the term and replaced it with the explanatory text.Lines 84-86

"Identified sites were searched for references/citations to other sites , which were then in turn searched for other references to sites that were not included in the initial searches."

  1. As the analysis was conducted by a medical student and psychiatrist and it’s not clear how this influenced the analytical process. Particularly, what measures were taken to ensure that people with lived experience were able to speak openly about their perspectives, and how were these perspectives privileged or valued?

Response: The analysis was conducted by a junior doctor (not medical student) and an academic psychiatrist. As both people carrying out the analysis were medically trained, it is possible that the thematic analysis may have been influenced by a medical bias in terms of developing themes. We have now acknowledge this in the text . The reviewer has raised an important question  about involvement of people with personal experience in the analysis – that is, in deciding what aspects of the data were useful/helpful and therefore worth pointing out.  We did not involve people with lived experience in the analysis and have added text to discussion acknowledging it as a limitation but pointing out that the workshop included nine people with personal experience. Lines 128-137

"As both people carrying out the analysis were medically trained, it is possible that the thematic analysis may have been influenced by a medical bias in terms of developing themes. To ensure intersubjective reproducibility and comprehensibility the analysis was regularly discussed with a multi-disciplinary team of professionals including self-harm experts (AH & CB), and qualitative researchers LB and KF. In addition, the results were reviewed and discussed at a one-day workshop which included eight people with lived experience of self-harm, two self-harm experts, three clinical psychologists, 2 psychiatrists, 1 junior doctor, 1 sociologist (KF) and 1 mental health nurse."

  1. The findings are listed on page 4, but the categories do not reflect this list. The presentation of findings is also inconsistent. Some sections of the findings are presented as a series of quotes and some are presented in tables. In both cases there is very little analysis, just descriptive quotes.

Response: We apologise if the text is not clear. We identified broad categories which were not necessarily related to group  content. These were based upon which audience the information was targeted at, or the type of information (personal account of self-harm) e.g. professionals, family or individuals who had self-harmed. Having reviewed the manuscript, we agree this is confusing and doesn’t really add to the overall findings. We have therefore removed this paragraph. We presented some of the findings in tables because they involved strategies which were listed on the sites to help people avoid imminent self-harm or ways to reduce thoughts of self-harm. We felt it would be helpful for a reader to see the different strategies and a table was the most efficient way of presenting the information. In terms of analysis, we felt that most of the quotes are self-explanatory and our purpose was to describe what information was available on the web that may be helpful.

  1. Finally, the discussion doesn’t bring the piece together. For example, the authors make the comment that some sites offered more practical advice, but it’s not clear how they came to this conclusion. There is little to no discussion about peer vs professional advice, which would seem an important consideration. The discussion needs to be more integrated with the wider literature to understand what this study adds, including what it adds to the previous 2001 study.

Response: We have considerably strengthened the discussion. We have changed the term practical to potentially helpful. We have included discussion of the role that clinicians may play in enquiring about Internet use. Lines 461-481

"Most information, however, is focused on cutting and there is still relatively little specific help that addresses self-poisoning. Much more information is available for specific demographic groups of people than in 2001 and there are many more personal accounts of recovery. A recent analysis of YouTube videos which described recovery from self-harm identified several themes which overlap with the ones we identified in this study: encouraging help seeking and talking about problems, substituting self-injurious practices by using alternative methods, a desire for change (motivation), and control over change (self-determination) [14].

The most detailed advice, however, was limited to a relatively small number of non-statutory sites, so a lay person or health professional who searches the web for help about self-harm may have to search through many different websites to find this useful guidance and help.

A recent study reported that almost one quarter of young adults in the community had accessed  Internet-content related to suicide/self-harm (15), and almost half of those who had self-harmed had searched for information about suicide or self-harm. A greater proportion of young people had accessed helpful sites than potentially harmful sites. Ten per cent of the whole sample of 3946 participants had searched for general information about self-harm, 8.2% for sites offering help, advice or support, and 9.1% for personal accounts of people who had hurt themselves. A smaller proportion (4%) had accessed sites dedicated to those who self-harm.

Another study of people attending hospital who had self-harmed also found high levels of internet use, particularly among children (16). It is therefore important that health professionals are aware of this, and feel comfortable discussing Internet use with the people they are trying to help. It has been suggested that clinicians should take an Internet history during psychosocial risk assessments for self-harm (17), and generally this seems to be something that clinicians feel is acceptable to do (16), although it is not known how common this is in clinical practice."

Reviewer 2 Report

I think this is a really interesting study looking at UK based resources for people who self-harm.  

I have some comments and suggestions that I hope the answers to which may further strengthen the ms.

  1. I think the introduction omits some relevant recent research (e.g. Marchant et al. 2017).
  2. Line 65, the authors describe the aim of the study as being to update the Prasad & Owens 2001 study and detail resources currently available on the internet for people who self-harm and their support networks.

If this is the case the ms would benefit from a) greater description of the Prasad & Owens study, b) an explanation as to why different search terms were included than the study, c) more reference/comparisons to the P&O study throughout the ms (next reference to the study is Line 323)

  1. Search terms: do the authors think that the search terms included reflect what would be searched by an ordinary individual? How were the terms selected? I understand the rationale for searching common terms for self-harm, but do the authors think anything was missed by not including terms like ‘overdose’/ suicide attempt/ behaviour (esp. as parasuicide included)?
  2. Line 218 splits across pages i.e ‘More general advice was provided’ is on p6 then continues on line 222 on P8.
  3. I think the discussion would benefit from a more critical discussion of the findings.
  4. Line 331 “Our findings provide a starting point for clinicians who wish to provide some guidance for their patients about internet help resources for self-harm” – could the authors clarify this statement?
  5. I think that it would improve the impact of the ms if the authors provided a more detailed section on the clinical implications and the important research questions that they believe should be addressed in future work.
  6. I get a sense of an overwhelming volume of available info. What/how authors think it could be made more accessible/ useable to individuals?

I hope the authors find these comments useful in revising their ms.

Author Response

  1. I think the introduction omits some relevant recent research (e.g. Marchant et al. 2017).

Response: We thank the reviewer for their comments. We have added the reference to Marchant et al. 2017). We had actually referred to the review but had omitted the reference. We apologise for this. Lines 52-56

“For example, a recent systematic review of the relationship between internet use, self-harm and suicidal behaviour in young people identified 46 studies of relevance [12]. The authors concluded there was the potential for internet use to reduce isolation and to be used as a source of help and therapy, and that clinicians who work or see people who self-harm should engage in discussion about internet use. “

  1. Line 65, the authors describe the aim of the study as being to update the Prasad & Owens 2001 study and detail resources currently available on the internet for people who self-harm and their support networks.If this is the case the ms would benefit from a) greater description of the Prasad & Owens study, b) an explanation as to why different search terms were included than the study, c) more reference/comparisons to the P&O study throughout the ms (next reference to the study is Line 323)

Response: We have added more details about the Prasad and Owens review in the introduction. It’s age indicates that there is nothing contemporary out there. We didn’t use exactly the same search terms as the Prasad and Owens review was more wide ranging and for instance looked at sites related to suicide. Lines 62-71

 “In the context of this debate, it would be useful for clinicians if guidance were available about which aspects of the internet they should discuss with patients, or where patients or their families and friends might be directed on the web to find help. Such guidance is however not readily found. We have been able to identify only one previous study, conducted in 2001, which sought to describe the information and help available on the internet for people who self-harm – aimed at informing clinicians about potential resources when discussing self-harm with their patients [14]. The authors  found only a small number of cites (31-37) with content about  various aspects of self-harm. Most of the cites provided constructive suggestions (e.g. how to overcome shameful feelings about self-harm, or advice on how to feel better about onself). Most of the information concerned self-injury and there was very little discussion of self-poisoning.”

  1. Search terms: do the authors think that the search terms included reflect what would be searched by an ordinary individual? How were the terms selected? I understand the rationale for searching common terms for self-harm, but do the authors think anything was missed by not including terms like ‘overdose’/ suicide attempt/ behaviour (esp. as parasuicide included)?

Response: The search terms were generated by a group discussion and by trying out some of the search terms to see what sites they found and  other common terms. We have added a comment in the discussion that some common terms such as overdose/ suicide attempt were omitted. Lines 351-354

“The search terms we used were generated within the research group by discussion and then tested out with preliminary searches to determine what sites they found or other terms were revealed. We realise that some other common terms such as ‘overdose’ were not included in the terms we generated.”

  1. Line 218 splits across pages i.e ‘More general advice was provided’ is on p6 then continues on line 222 on P8.

‘More general advice was provided’ is just split by Table 2 so the sentence continues between Table 2 and Table 3. Now line 231 and 235

  1. I think the discussion would benefit from a more critical discussion of the findings.

  1. Line 331 “Our findings provide a starting point for clinicians who wish to provide some guidance for their patients about internet help resources for self-harm” – could the authors clarify this statement?
  2. I think that it would improve the impact of the ms if the authors provided a more detailed section on the clinical implications and the important research questions that they believe should be addressed in future work.
  3. I get a sense of an overwhelming volume of available info. What/how authors think it could be made more accessible/ useable to individuals?

Response to above 3 points. We have added further discussion regarding clinical implications  and in doing so clarified the point about ‘clinicians wishing to provide guidance. Lines 366-375

"Enquiry about internet use should be included in clinical assessments of people at risk of self-harm, especially young people. Royal College of Psychiatrists guidance to clinicians emphasises the need to explore possible harms [17]: a fuller discussion should be supplemented with questions about what has been found helpful and with advice about sites or organisations that can be readily accessed. [16]. Such advice should always come with a comment that what is helpful varies by individual, and with an offer to discuss experiences again if wanted.

Enquiry about internet use should be included in clinical assessments of people at risk of self-harm, especially young people. Royal College of Psychiatrists guidance to clinicians emphasises the need to explore possible harms [17]: a fuller discussion should be supplemented with questions about what has been found helpful and with advice about sites or organisations that can be readily accessed. [16]. Such advice should always come with a comment that what is helpful varies by individual, and with an offer to discuss experiences again if wanted.

Most of the advice that we found was quite basic and may be helpful for  people with little or no personal experience of self-harm but is unlikely to be helpful for people who have been self-harming for some time. Any discussion or reference to particular sites should not come with a recommendation that they are good, but more that the person may find them of some use."

Reviewer 3 Report

Overall, I loved this topic and appreciate the careful approach. I recommend the authors include studies done on pro-anorexia websites which draw similar conclusions, and use (sometimes) similar methodologies, e.g. Balter-Reitz, S., & Keller, S. (2005). Censoring thinspiration: the debate over pro-anorexic web sites. Free Speech Yearbook42(1), 79-90.

In terms of the presentation of methods, I had some questions. For example, it was difficult to determine how the 49 targeted websites led to the 250+ targeted websites that were included in the total sample of 350+. This need to be clarified. It was also hard to tell why the authors chose their exclusion criteria. For e.g., why would you exclude websites that require log-in or registration? It would seem to me that these are the very ones that individuals at risk of self-harm might use to disclose more personal concerns, support, or risk-promotion. Similarly, with online forums. Why exclude?

Second, I was confused as to how the authors developed their coding categories. Why were no risk-promotion categories included? If inductive thematic analysis was used, I would predict that at least some of the material would encourage risk, or provide practical advice on self-harm strategies. Yet, none of the results presented include such content or categories -- at least not that I could find.

I recommend that the content be re-analyzed for any possible destructive messages. If none are found, then the authors need to explain why they think that might be. 

Author Response

1. Overall, I loved this topic and appreciate the careful approach. I recommend the authors include studies done on pro-anorexia websites which draw similar conclusions, and use (sometimes) similar methodologies, e.g. Balter-Reitz, S., & Keller, S. (2005). Censoring thinspiration: the debate over pro-anorexic web sites. Free Speech Yearbook42(1), 79-90.

Response:We thank the reviewer for their comments. We considered including refs to some of the debates around pro-anorexic websites, and we thank the reviewer for their suggestion, however, we felt it might too confusing to reference another area of 'heated debate' re Internet use, as our research focus is not eating disorders.

2. In terms of the presentation of methods, I had some questions. For example, it was difficult to determine how the 49 targeted websites led to the 250+ targeted websites that were included in the total sample of 350+.  

Response: We apologise for the lack of clarity. We have added further explanation to the text. Pages 147-153

The search strategy, number of results and duplicates, and details of results included for each constituent website can be found in Appendix B. Each of the 49 websites are listed with the number of hits for self-harm, duplicates and the number of hits  that met eligibility criteria for full text inclusion e.g. the website for the National Institute for Health and Care Excellence  (NICE) had 82 hits (16 duplicates) of which 33 met eligibility for full text inclusion. The 49 sites generated 2,694 hits (excluding duplicates) of which 383 hits met eligibility for full text inclusion. 

3. This need to be clarified. It was also hard to tell why the authors chose their exclusion criteria. For e.g., why would you exclude websites that require log-in or registration? It would seem to me that these are the very ones that individuals at risk of self-harm might use to disclose more personal concerns, support, or risk-promotion. Similarly, with online forums. Why exclude?

Response: We didn’t want to review all online content.  Our focus was specifically on sites that might contain direct and easily accessible guidance. We already know from previous work that interactive sites contain such a bewildering amount of material that finding advice is too laborious, and families or health professionals are unlikely to access chat rooms.

4. Second, I was confused as to how the authors developed their coding categories. Why were no risk-promotion categories included? If inductive thematic analysis was used, I would predict that at least some of the material would encourage risk, or provide practical advice on self-harm strategies. Yet, none of the results presented include such content or categories -- at least not that I could find.I recommend that the content be re-analyzed for any possible destructive messages. If none are found, then the authors need to explain why they think that might be. 

Response: Our purpose was to find potentially helpful content on-line, so the above point was outside our remit.  We have clarified this in the introduction section.Lines 76-78

"The aim of the present study was to investigate, characterise and describe the resources currently available on the internet for people who self-harm, or their friends or families, which may be potentially helpful. We focused on UK-generated sites, so that our findings could be used by UK clinicians to guide their own searches and subsequent discussions with their patients."